# Global meta-analysis reveals overall higher nocturnal than diurnal activity in insect communities

Mark K. L. Wong [1,2] ✉ & Raphael K. Didham [1,2]

Insects sustain key ecosystem functions, but how their activity varies across the day–night cycle and the underlying drivers are poorly understood. Although entomologists generally expect that more insects are active at night, this notion has not been tested with empirical data at the global scale. Here, we assemble 331 quantitative comparisons of the abundances of insects between day and night periods from 78 studies worldwide and use multi-level meta-analytical models to show that insect activity is on average 31.4% (CI: −6.3%–84.3%) higher at night than in the day. We reveal diel preferences of major insect taxa, and observe higher nocturnal activity in aquatic taxa than in terrestrial ones, as well as in warmer environments. In a separate analysis of the small subset of studies quantifying diel patterns in taxonomic richness (31 comparisons from 13 studies), we detect preliminary evidence of higher nocturnal richness in tropical than temperate communities. The higher overall (but variable) nocturnal activity in insect communities underscores the need to address threats such as light pollution and climate warming that may disproportionately impact nocturnal insects.

Insects are of tremendous ecological and economic importance because they sustain food webs, pollination networks, and the functioning of ecosystems globally[1,2]. However, many insects are currently at risk from compounding threats from human activities such as deforestation, agricultural intensification and urbanisation[3]. Despite the urgent need to understand and conserve insect biodiversity, basic knowledge on insect distributions and ecology is in short supply[4]. In particular, the fundamental rhythms in the abundance and diversity of active insects across the day–night (diel) cycle are poorly documented for most ecosystems. The factors shaping diel patterns in insect activity are likewise unclear[5].

There is a need to better understand the diel activity patterns of insect communities because these patterns influence the exposure of insects to environmental changes as well as their associated ecosystem functions considerably. Especially for ectothermic organisms, activity periods reflect environmental constraints on the times during which the interactions essential for population persistence such as resource acquisition, dispersal, and reproduction can occur[6]. Moreover, insect activity drives many crucial ecosystem functions such as herbivory, pollination, the predation of and by other organisms, and ecological engineering[7]. Investigating the diel activity patterns of insect communities is therefore not only vital for comprehending the temporal mechanisms determining community structure but also the daily rhythms in key ecosystem functions. Such pursuits are timely considering mounting evidence of anthropogenic threats, such as light pollution, affecting specific communities and functions[8,9].

Most studies on the temporal dynamics of insect communities have examined changes in insect abundance and diversity at intermediate to long timescales[10,11], such as those corresponding to annual climatic seasons[12,13], the phenological flowering windows of host plants[14,15], or timing of natural and anthropogenic disturbances[16,17]. In contrast, ecologists lack an understanding of how and why insect biodiversity varies at finer timescales, most notably across the 24-h diel cycle[18]. While anecdotal accounts have suggested that most insects are

[1]School of Biological Sciences, The University of Western Australia, Crawley, WA 6009, Australia. [2]CSIRO Health & Biosecurity, Centre for Environment and Life Sciences, Floreat, WA 6014, Australia. ✉e-mail: mark.wong@uwa.edu.au

nocturnal[19], empirical evidence has been mixed, with some recording greater numbers of insects in the day[18] and others at night[20].

Methodological shortcomings largely account for the slow progress in documenting the diel dynamics of insect communities. The logistical obstacles hindering diurnal ecologists from effectively studying animals at night are nontrivial[21]. Moreover, many 'standard' sampling techniques for collecting insects are unsuitable for investigations across the entire spectrum of diel activity because they inherently vary in collection efficiency between day and night (e.g. coloured pan-traps during the day, or light traps at night), or they inadvertently capture inactive individuals (e.g. sweep-netting, litter sampling, beating vegetation) (Fig. 1); noting that some of these have nevertheless been used in earlier diel comparisons (e.g. ref. 22). Still, there exist several methods that exclusively collect active individuals and can provide comparable collections across diel periods, such as movement-based interception traps (e.g. pitfall traps, malaise traps, drift nets) and some attraction-based bait traps (e.g. dung-baited pitfall traps) (Fig. 1).

To obtain a global overview of diel variation in abundance and richness in insect communities, we collected data from published studies that systematically sampled insect communities using suitable and comparable collection methods in both day and night periods. We then performed multi-level meta-analyses to quantify the diel variation in the abundance of insects and the numbers of different taxa in communities, and to elucidate potential environmental moderators of such variation (detailed in Methods). Our findings show that while insect activity is generally higher during the night, diel patterns in insect activity vary extensively across the Earth's surface, reflecting the effects of various abiotic and biotic mechanisms that remain poorly understood.

## Results

We identified 99 studies published between 1959 and 2022, which provided 386 observations of diel patterns in abundance and/or taxonomic richness in insect communities. The studies spanned all continents except Antarctica and encompassed a wide range of habitats, including both terrestrial and aquatic systems. Nevertheless, as with most global biodiversity assessments[23], tropical regions remained underrepresented relative to the high concentrations of biodiversity in these areas[24,25]. The insects sampled across all studies exceeded 3 million individuals from 16 taxonomic orders, and the number of species in a community ranged from 1 to 326 ($M = 32.2$).

### Generally higher insect abundance at night

Using multi-level meta-analysis modelling, the abundance of insects was found to be higher in the night than in the day by an average of 31.4%, but this relationship varied considerably (CI: −6.3–84.3%) across the 331 observations from 78 publications in which it was possible to calculate an effect size (intercept-only model, 'RE.null', conditional $R^2 = 0.62$, AIC = 1088.2; Table 1) (Fig. 2). There was high heterogeneity in the effect of diel period (i.e. night vs. day) on insect abundance both within and between publications (random variance components in RE.null model, $I^2_{Total} = 89.3$, $I^2_{Study} = 55.7$, $I^2_{Residual} = 33.6$).

Of 18 moderators tested−including various environmental (e.g. climate, latitude, elevation, habitat), disturbance-related (integrated human pressures and artificial sky luminance), methodological (collecting methods) and taxonomic factors (Supplementary Table 1)− eight single-moderator models outperformed the intercept-only model (based on the Akaike information criterion, 'AIC') and explained significant variance in the heterogeneity of effect sizes (Table 1).

### Macroecology of diel abundance patterns

Heterogeneity in the effect of diel period on insect abundance was strongly influenced by the taxonomic composition of the community (single-moderator model 'RE.taxa', ΔAIC = −16.3, marginal $R^2 = 0.16$) (Fig. 3), with multiple taxonomic groups showing significantly higher abundance during a specific diel period. The abundances of mayflies (Ephemeroptera), caddisflies (Trichoptera), moths and butterflies (Lepidoptera) and earwigs (Dermaptera) were

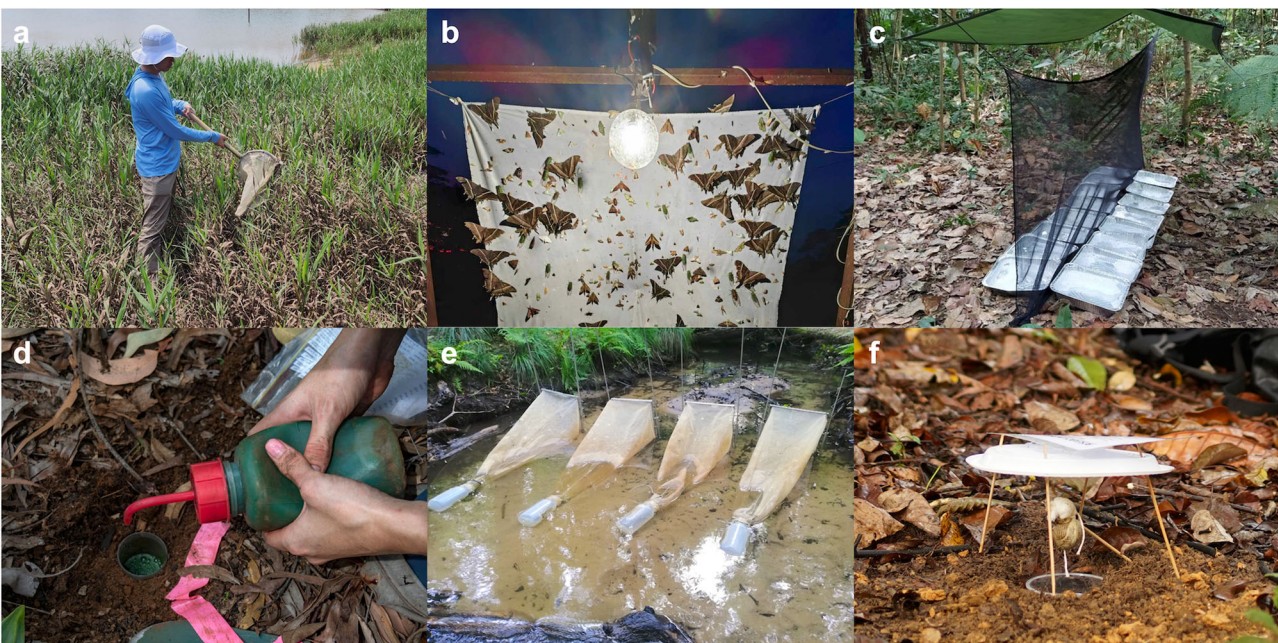

**Fig. 1 | Common methods for sampling insects vary in their capacities to elucidate diel activity patterns.** Sweep-netting (**a**) results in the capture of individuals that may be inactive during the sampling period, while light traps (**b**) inherently vary in collection efficiency between day and night, and are not suitable for unbiased diel comparisons. By contrast, sampling methods that intercept moving insects such as flight-interception traps (**c**), pitfall traps (**d**) and drift nets (**e**), as well as sampling methods using non-visual attractants such as food baits (**f**) provide comparatively unbiased comparisons of insect activity between day and night periods. Photographs courtesy of Roger Lee (**a**), Nicky Bay (**b**), the TEE Lab of the Asian School of the Environment (**c**, **f**), Francois Brassard (**d**) and Sebastian Prati (**e**).

**Table 1 | Summary of multi-level meta-analytical models for the effect of diel period on insect abundance**

| Model | Moderator(s) | AIC | Marginal $R^2$ | Conditional $R^2$ | $I^2_{Total}$ | $I^2_{Publication}$ | $I^2_{Effect\ size}$ |
|---|---|---|---|---|---|---|---|
| RE.multimod | taxonomic group, ecosystem, maximum temperature, average precipitation | 1047.98 | 0.34 | 0.76 | 86.89 | 55.73 | 31.17 |
| RE.taxa | taxonomic group (18 levels) | 1071.92 | 0.16 | 0.71 | 88.51 | 57.95 | 30.56 |
| RE.sampling | sampling method (3 levels: attraction, movement.interception, other) | 1074.31 | 0.12 | 0.69 | 89.66 | 58.48 | 31.18 |
| RE.habitat | habitat (6 levels: terrestrial.other, forest, grassland/savanna, aquatic.other, stream, river) | 1076.10 | 0.24 | 0.65 | 87.17 | 47.10 | 40.07 |
| RE.ecosystem | ecosystem (2 levels: aquatic, terrestrial) | 1077.37 | 0.19 | 0.64 | 87.70 | 48.72 | 38.98 |
| RE.clim.prec | average precipitation | 1078.22 | 0.02 | 0.58 | 91.36 | 51.67 | 39.69 |
| RE.npp | average net primary productivity | 1081.00 | 0.06 | 0.72 | 90.61 | 63.51 | 27.10 |
| RE.clim.tmax | maximum air temperature | 1081.50 | 0.06 | 0.65 | 89.17 | 56.46 | 32.71 |
| RE.elevation | elevation | 1084.03 | 0.05 | 0.65 | 89.25 | 55.93 | 33.32 |
| RE.null | null (intercept-only model) | 1088.20 | 0.00 | 0.62 | 89.33 | 55.71 | 33.62 |

All models included a publication identifier and an effect size identifier as random effects. Presented – in order of ascending AIC value – are the details of the intercept-only model which only included random effects (model 'RE.null'), the eight single-moderator models which comparatively outperformed the intercept-only model, and the full multi-moderator model (model 'RE.multimod'). AIC values were calculated from models estimated with maximum likelihood (for facilitating model comparisons), while $R^2$ and $I^2$ values were calculated from models estimated with restricted maximum likelihood (for reporting). Details of the full set of models analysed are provided in Supplementary Table 1.

significantly higher during the night, whereas thrips (Thysanoptera), and bees and wasps (Hymenoptera) were significantly more abundant during the day. There were also weaker, non-significant, tendencies for the two large insect orders, beetles (Coleoptera) and flies (Diptera) to be more numerous on average at night, whereas ants (Hymenoptera: Formicidae), booklice (Psocoptera) and grasshoppers, crickets and katydids (Orthoptera) tended to be more numerous during the day, on average, but with high variation in diel preference across studies.

Heterogeneity in the effect of diel period on insect abundance was also strongly influenced by habitat type (single-moderator model 'RE.habitat', ΔAIC = −12.1, marginal $R^2$ = 0.24) (Fig. 4), with clear differences between aquatic and terrestrial habitat types. Insect abundance was higher during the night by an average of 242.5% in rivers (CI: 47.7%–694.1%, $P < 0.01$) and 115.3% in streams (CI: 34.4%–245%, $P < 0.01$). By contrast, in grasslands and savannas, insect abundance was lower during the night by an average of 75.2% (CI: −90.3% to −37.1%, $P < 0.01$). In forests, insect abundance was on average 11.8% lower during the night (CI: −44.1–39.3%), but not significantly so ($P = 0.59$).

The effect of diel period on insect abundance was also shaped by distinct topographic, climatic and productivity gradients (Fig. 5; single-moderator models 'RE.clim.tmax', 'RE.clim.prec', 'RE.elevation' and 'RE.npp' in Table 1). Overall, higher insect abundance during the night was observed in environments with higher maximum temperatures and higher average precipitation as well as areas of lower productivity and lower elevation (Fig. 5).

Combining these effects and accounting for potential collinearity among predictors, the best multiple-moderator model (model 'RE.multimod', ΔAIC = −40.2, marginal $R^2$ = 0.34) included the additive effects of the taxonomic composition of the community, the predominant ecosystem type (terrestrial vs. aquatic systems), as well as the maximum temperature and average precipitation in the environment. The effects of these moderators in the multiple-moderator model were consistent with their effects in the single-moderator models. Specifically, higher daytime insect abundance was significantly associated with communities containing thrips, ants, bees or wasps, while higher nighttime abundance was significantly associated with communities in aquatic ecosystems exposed to higher maximum temperatures and higher precipitation.

In the analysis, single-moderator models for the effects of human disturbances in the surrounding environment such as the amount of artificial sky luminance ('ALAN'; data from ref. 26) and the magnitude of integrated human pressures (measured by the Human Footprint Index, 'HFP'; ref. 27) did not outperform the intercept-only model (Supplementary Table 1).

## Diel patterns in insect richness

Likely owing to taxonomic impediments, surprisingly few studies reported the mean richness of insect taxa (i.e. mean number of different taxa) observed during a specific diel period, which was required to calculate an effect size. Only 31 such comparisons were available from 13 publications; insects were identified to species in 30 comparisons and to families in one. Across these studies, the total number of taxa in a community ranged from 19 to 108 ($M = 51$). A multi-level meta-analysis for this limited dataset showed that observed richness was on average 27.7% lower in the night, but not significantly so (CI: −55.2–16.8%) (intercept-only model, 'rich.RE.null', AIC = 25.6; Supplementary Table 2). These preliminary results should be interpreted with caution, given the limited nature of the data. There was also very high heterogeneity in the effect of diel period on richness between publications (from rich.RE.null, $I^2_{Total}$ = 99.9, $I^2_{Study}$ = 98.7, $I^2_{Residual}$ = 1.22). Interestingly, the best model for the effect of diel period on richness included the effect of absolute latitude (model 'rich.RE.lat', marginal $R^2$ = 0.39, ΔAIC = −7.76; Supplementary Table 2), whereby an increasing distance from the equator was associated with a lower richness of insect taxa observed at night (Supplementary Fig. 1).

## Discussion

Our results put hard empirical data behind the widely held truism that there are generally more insects out at night[19]. They also show that the differences in insect numbers between day and night periods can vary extensively across regions on Earth. Moreover, in uncovering multiple key moderators of this variation—which include a range of abiotic factors (climate, elevation) and biotic factors (e.g. habitat type, taxonomic constitution, productivity gradients)—our results underscore that the mechanisms governing the diel dynamics of insect community structure are likely rich and varied. Below, we discuss a few of these potential mechanisms (e.g. abiotic regulation, predator avoidance, resource tracking, intraspecific variation), noting however that others are likely also at play, and their specific roles in structuring the diel dynamics of insect communities will best be illuminated through additional field and experimental studies across the 24-h diel cycle. We emphasise the urgency for such work in the face of growing anthropogenic threats that may disproportionately impact specific diel communities.

The effects of temperature and elevation in shaping the distribution of insect abundance across day and night periods in

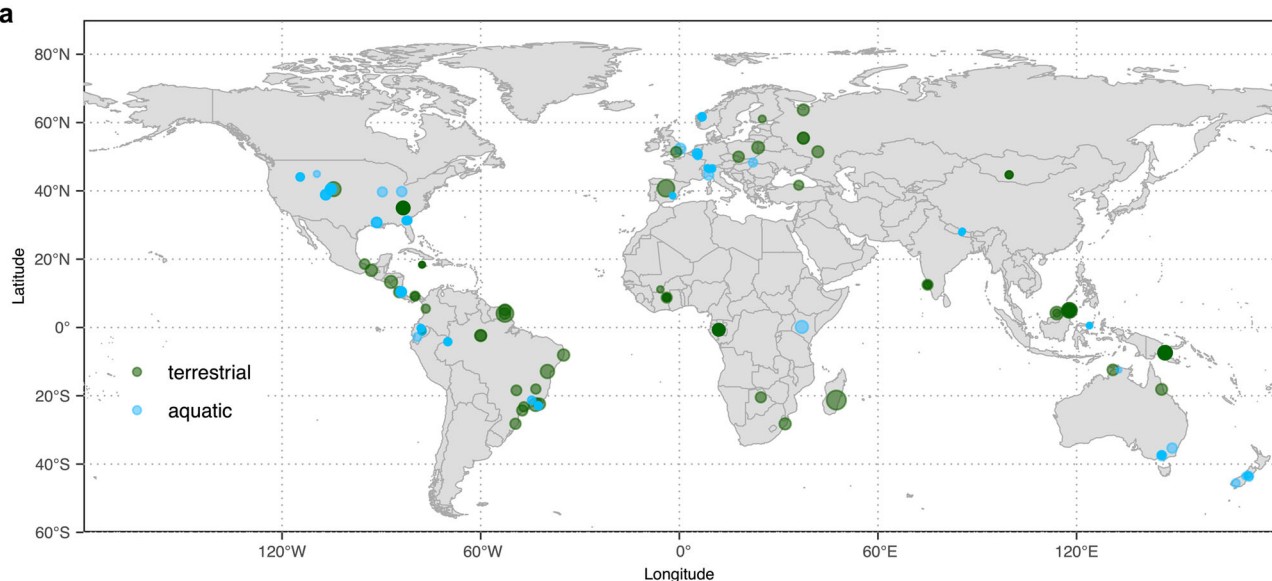

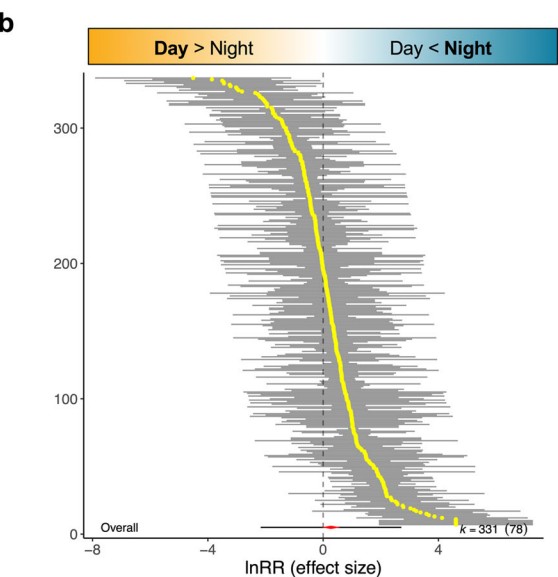

**Fig. 2 | Diel patterns in insect abundance globally. a** Distribution map of 331 observations of diel patterns in insect abundance from 78 studies in terrestrial (green) and aquatic (blue) ecosystems. The opacity of each point corresponds to the number of observations for a given locality. The size of each point corresponds to the relative sampling effort (measured as the number of samples collected) in each observation. **b** Caterpillar plot for the intercept-only model (model 'RE.null') for the effect of diel period (night vs. day) on the abundance of active insects in a community. Plot shows the individual effect sizes (log response ratio, lnRR) from

the 331 day–night comparisons and their associated confidence intervals. Increasingly negative and positive effect-size values correspond to higher insect abundance in the day and night, respectively, as illustrated at the top of the plot. The overall mean effect size is centred in the red diamond at the bottom of the plot, and equates to a 31.4% increase in the abundance of active insects during the night. The associated 95% confidence intervals (CI:-6.3%–84.3%) for the mean effect extend horizontally to the ends of the red diamond while the prediction intervals (black lines) extend from the red diamond.

communities worldwide are consistent with the notion that the activities of these ectothermic organisms are broadly thermally constrained[6]. Thermal Performance Theory[28] posits that the physiological performance of ectotherms increases with temperature until reaching a peak, beyond which there is a rapid decline in physiological performance, ultimately leading to mortality. Given that environmental temperatures peak during the day, higher maximum environmental temperatures may select for increased nocturnality in insect communities (Fig. 5a) as more individuals avoid heat stress from daytime temperatures that approximate their upper thermal limits[29]. Conversely, with increasing elevation (Fig. 5d), lower temperatures during the night may constitute a selection pressure on insects' lower thermal limits[30], thereby promoting increased diurnality.

Nonetheless, although abiotic regulation has traditionally been viewed as the principal mechanism determining the diel activities of insects[31], our results suggest that other processes are also relevant. In particular, the contrasting diel patterns of insect abundance in aquatic and terrestrial habitats contribute to a growing appreciation for the differing dynamics of ecological communities on land and in water[32] and may result from distinct mechanisms. Across studies of aquatic insects in our analysis, night-time drift was widely regarded as a strategy for reducing the risk of predation from visually hunting fishes[33–35]. Evidencing this hypothesis, aperiodic insect drift was observed in streams historically devoid of fishes, while a high nocturnal drift density was often observed in streams where fish occurred[33,34,36,37]. Most compellingly, introducing trout to

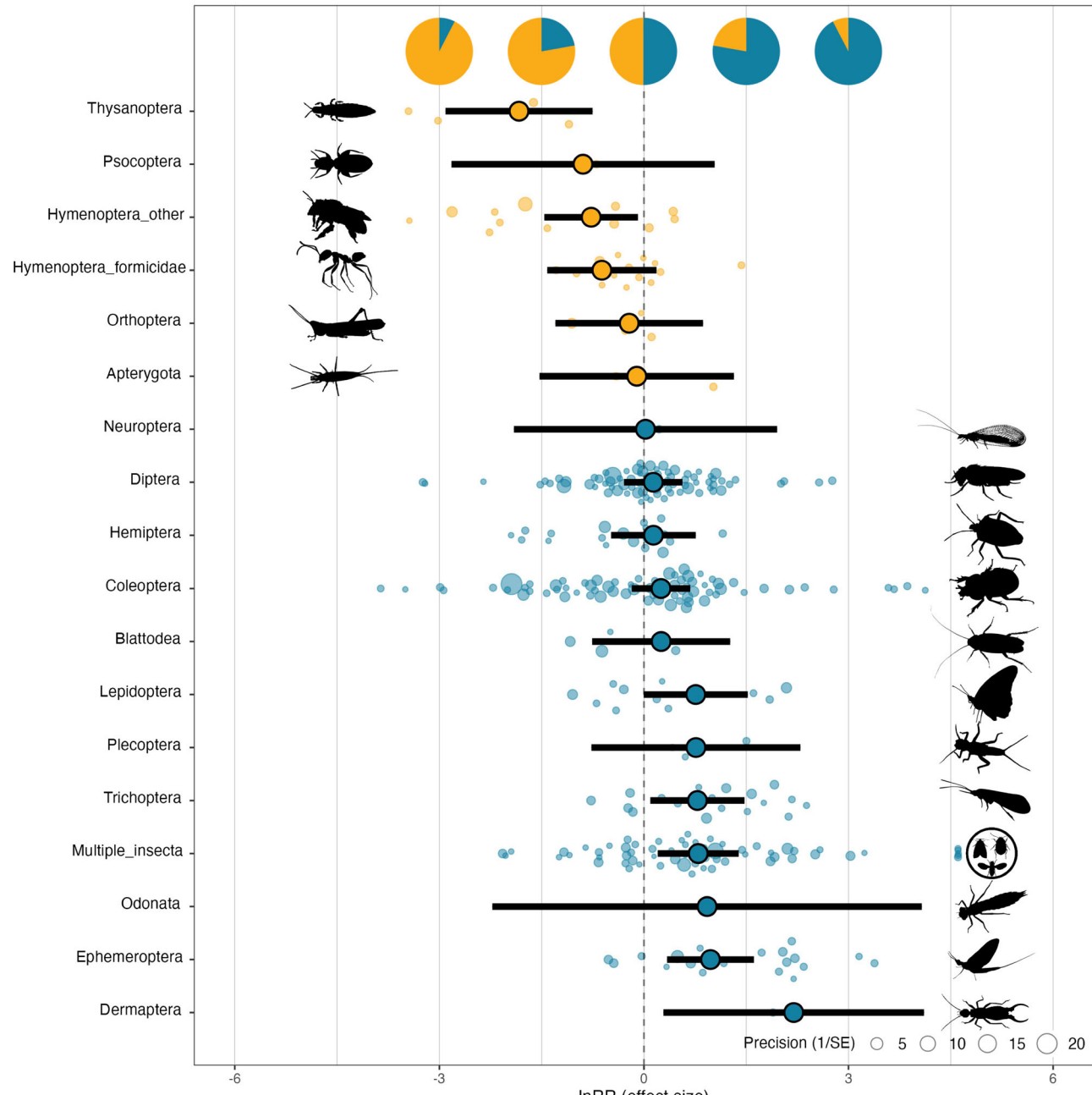

**Fig. 3 | Diel patterns in abundance vary among insect taxa ($k$ = 331 effect sizes).** For each of 18 taxonomic groups, the plot (from the single-moderator model, 'RE.taxa') shows the mean log response ratio (lnRR) estimate (encircled dot), 95% confidence intervals (bold line), the individual effect sizes (faded dots) and their precision (inverse standard error, 1/SE) for the effect of diel period on abundance. Taxa are presented in the order of ascending mean effect sizes from top to bottom.

Taxa associated with higher abundance in the day and night are indicated in orange and blue, respectively. The horizontal positions of pie charts correspond to effect size values, with each chart illustrating the relative proportions of individuals in a community that would be active in the day (orange) and night (teal) for a given effect size.

fish-free streams increased diel periodicity in the drift of baetid mayflies[36].

In contrast, the avoidance of visual predators would poorly explain the diel activity patterns of insects in terrestrial habitats, where higher insect activity was generally observed during the day. Moreover, diurnal insect activity was highest in open habitats—grasslands and savannas (Fig. 4)—where insects would conceivably be most exposed to visual predators. Nocturnal predators using non-visual hunting strategies (e.g. bats) may therefore play a more dominant role in shaping the diel activity patterns of terrestrial insect communities[38]. However, this notion and the generally higher insect activity observed

in the day across terrestrial habitats (Fig. 4) counters hypotheses by refs. 39,40 that insect activity in forests should be highest at night when herbivorous insects escape from day-active predators and exploit the accumulation of the day's photosynthate before it is translocated or respired. Interestingly, their hypotheses also find limited support in the apparent decrease in nocturnality with increasing NPP (Fig. 5c).

More broadly, given the sheer ecological diversity of insects on land, it is likely that a range of other biotic mechanisms besides predation determine their diel activity (competition and resource partitioning, resource tracking, mutualisms etc.). In this regard, the high

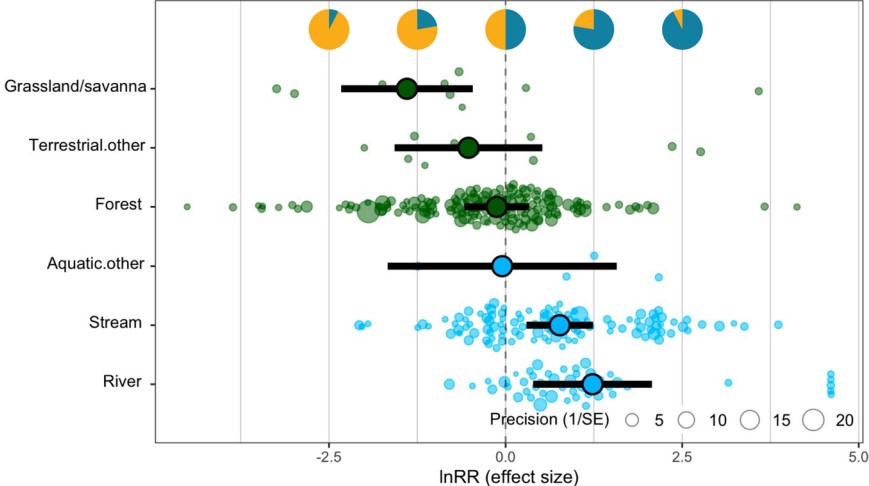

**Fig. 4 | Diel patterns in insect abundance vary across habitat types ($k = 331$ effect sizes).** For each of six habitat types, the plot (from the single-moderator model, 'RE.habitat') shows the mean estimate (encircled dot), 95% confidence intervals (bold line), the individual effect sizes (faded dots) and their precision (inverse standard error, 1/SE) for the effect of diel period on insect abundance. Habitat types are presented in the order of ascending mean effect sizes from top to bottom. Terrestrial and aquatic habitats are indicated in green and blue, respectively. The horizontal positions of pie charts correspond to effect size values, with each chart illustrating the relative proportions of individuals in a community that would be active in the day (orange) and night (teal) for a given effect size.

diversity and compositional variation of insect communities in forests[25] likely accounts for the extensive heterogeneity in diel patterns observed here (Fig. 4). Indeed, taxonomic composition alone explained a substantial amount of the variation (16%) across diel patterns of insect communities globally (Fig. 3), suggesting that the diel activities of different taxa and trophic groups (consumers, predators, parasites etc.) are possibly regulated by different mechanisms.

It is also possible that the variation in diel activity patterns observed across insect communities is shaped to some degree by intraspecific variation in the activity patterns of individuals from different stages across the life cycle. Although some studies in our analysis only sampled adults (e.g. studies on ants, wasps and ground beetles), others targeted juveniles (e.g. studies on the drift of chironomid larvae in streams), and many used methods that may have collected individuals from different life stages without distinguishing these in samples (e.g. pitfall traps and drift nets may collect both juvenile and adult insects in terrestrial and aquatic ecosystems, respectively). Yet in many insects, juveniles differ markedly from adults in ways that should influence their periods of activity across the diel cycle, such as in their specific diets and habitats, mobility and foraging patterns, and susceptibility to predators[41,42]. Moreover, juvenile and adult insects can possess different thermal sensitivities, with the thermal tolerances of some species either increasing or decreasing following developmental change[43]. While we suspect that nocturnal insect communities in aquatic and terrestrial systems may be disproportionately comprised of juvenile individuals avoiding predators or higher temperatures, this remains to be tested empirically.

Although the amount of artificial sky luminance (ALAN)[26] and the magnitude of integrated human pressures (HFP)[27] in the surrounding environment did not strongly influence the documented diel patterns in insect abundance in our analysis, this was likely because most of the studied insect communities were situated in areas less affected by human activities and only exposed to a limited range of values in these two moderators ($M_{ALAN} = 0.12 \pm 0.35$ mcd/m$^2$ vs. global maximum of >7.3 mcd/m$^2$; $M_{HFP} = 8.67 \pm 8.43$ vs. global maximum of 100). Still, given that the coefficients of both moderators were negative (Supplementary Table 1), increasing artificial sky luminance and human pressures could potentially be associated with lower abundances of

insects at night. To this end, additional studies on the diel activity patterns of insect populations and communities along disturbance gradients may clarify the mechanisms driving human impacts on insect biodiversity.

In showing that insect abundance varies across the diel cycle, our findings shed light on a general but still poorly understood ecological phenomenon. In particular, the diel partitioning of species richness in insect communities is largely unexplored. It remains to be seen whether diel patterns in species richness mirror those in abundance across insect communities globally, or if these indeed track other environmental gradients, such as the preliminary pattern we detected along the latitudinal gradient (see Supplementary Fig. 1). To this end, increasingly accessible molecular methods for identifying specimens such as DNA barcoding[44] offer promising avenues for overcoming the taxonomic impediments that have hindered studies on the diel dynamics of insect communities in diverse regions, particularly tropical ecosystems. In addition to clarifying the mechanisms structuring diel patterns in insect richness and abundance across spatial scales, there is much room for exploring the diel dynamics of insect communities at higher temporal resolution (e.g. including crepuscular periods), and investigating how diel shifts in insect community structure shape temporal fluctuations in ecosystem functions[45,46] via ecological theory[47] and trait-based approaches[48].

Concerted efforts to elucidate the diel dynamics of insect communities and their associated ecosystem functions are needed especially given the compounding anthropogenic threats to insects globally[3]. In particular, our finding of high nocturnal activity in many insect communities worldwide strongly underscores the urgency of addressing human activities that disproportionately impact nocturnal communities. Besides the burgeoning evidence for the detrimental effects of light and noise pollution on nocturnal insects[19,49,50], climate warming may further shorten periods of biophysically feasible activity for nocturnal insects in warm environments such as tropical forests, with complex consequences for fitness such as accelerated metabolic costs combined with reduced food intake[51,52]. As insects are among the most diverse and important organisms on Earth, studying their intricate rhythms with the rise and setting of the sun represents not just a scientific endeavour, but an imperative for preserving biodiversity in the Anthropocene.

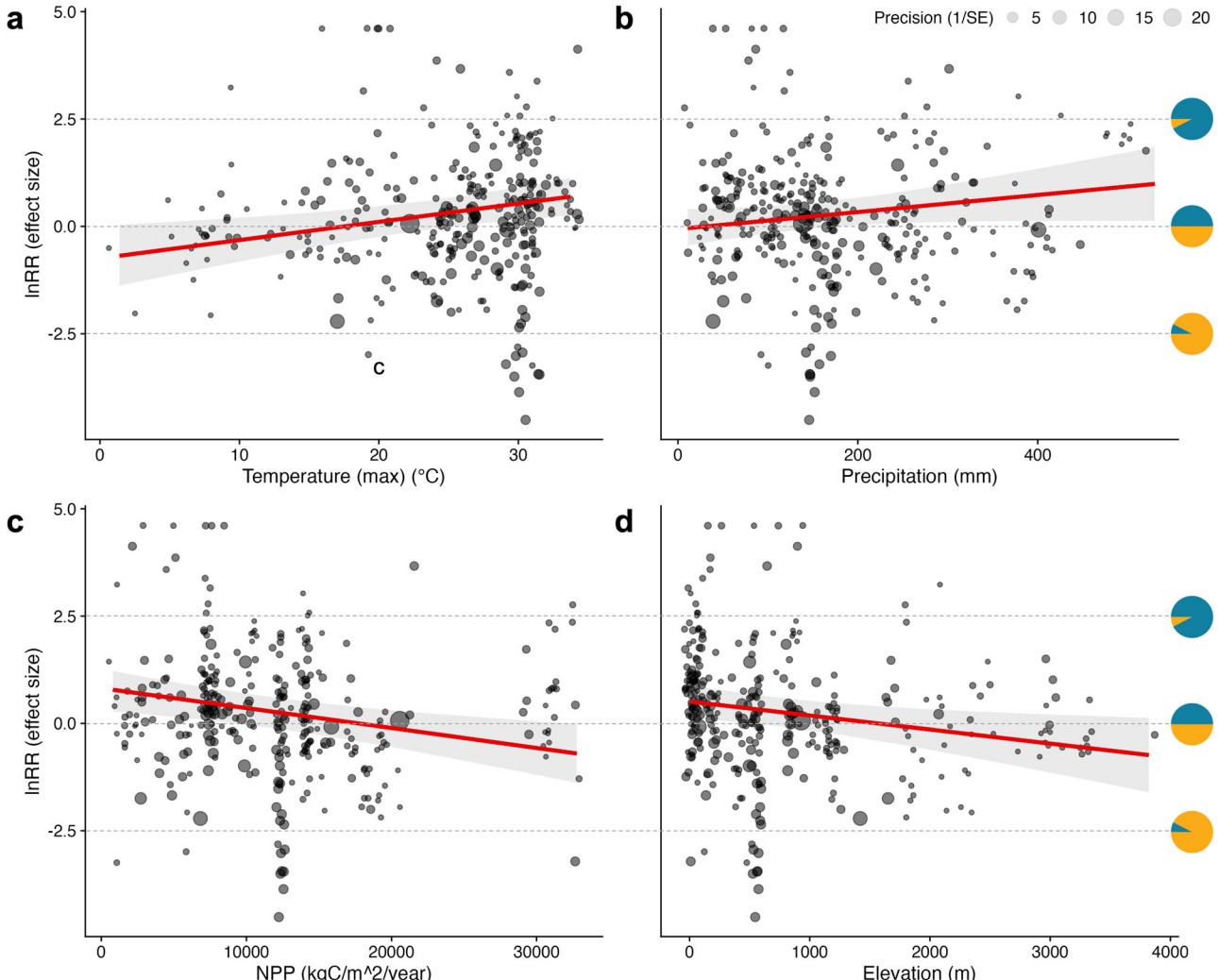

**Fig. 5 | Environmental gradients shape diel patterns in insect abundance.** Plots show the distribution of 331 effect sizes along four environmental gradients, where increasingly negative and positive effect sizes correspond to higher insect abundance in the day and night, respectively. The vertical positions of pie charts correspond to effect size values, with each chart illustrating the relative proportions of individuals in a community that would be active in the day (orange) and night (teal) for a given effect size. Shaded ribbons indicate the 95% confidence intervals of the mean. In general, higher night-time abundances are observed in environments with higher maximum temperatures (**a**), higher average precipitation (**b**), lower net primary productivity (**c**) and lower elevation (**d**).

## Methods
### Identifying relevant publications
We performed a literature search in all Web of Science databases on 28th April 2022 for studies that sampled insect communities across the diel cycle. The search terms used were '(insect) AND (community OR communities) AND (activity OR diel OR nocturnal OR diurnal OR night OR day)'. We sorted the 19,983 results by relevance and manually screened the abstracts of the first 2000 results to identify relevant publications (beyond the 1350th result, zero to two relevant publications were identified out of every 50 results, while no relevant publications were identified beyond the 1850th result). We only included a study in our meta-analysis if it systematically sampled an insect community using the same collection method during the day and the night, and separated the samples collected from each diel period. The two diel periods were consistently defined across studies; 'day' referred to the period after sunrise and before sunset, while 'night' referred to the period after sunset and before sunrise. We only included studies that used sampling methods that would collect active individuals, such as movement-based interception traps (e.g. pitfall traps, sticky traps, malaise traps, drift nets) and some attraction-based traps (e.g. dung-baited pitfall traps). We excluded studies that used methods which could potentially collect inactive individuals (e.g. sweep-netting, beating) as well as methods for which collection efficiency or attractiveness was influenced by environmental changes across the diel cycle, such as light traps and coloured pan-traps.

### Data compilation
For each relevant study, we recorded the mean abundance of individuals (and where reported, the mean numbers of taxa) in each day and night sample, as well as the corresponding standard deviations (SD) and sample sizes ($n$). In most studies, each sample was a single collection unit (e.g. a single trap or net). We also recorded the total abundance of individuals (and where reported, the total numbers of each different taxon) across all day and night samples combined. Most studies measured insect abundance in terms of the numbers of individuals encountered (a minority used frequencies of occurrence or biomass), and where insect taxa were identified, this was often to the species level (with a few identifying to the genus, subfamily or family level).

To obtain data on potential factors shaping patterns in insect abundance and richness across the diel cycle, we recorded information on the geographic location, habitat, sampling period,

sampling method and sampled taxa in each study (Supplementary Table 4). Using the reported geographic coordinates of the sampled localities, we also determined the surrounding elevation (if not reported within the study) from Google Earth, and obtained values for multiple environmental indicators within a buffer radius of 1000 m from the sampled locality: the average, minimum and maximum values of air temperature as well as the average precipitation and solar radiation over the month(s) of sampling from WorldClim[53]; the average value of Net Primary Productivity (NPP) from the MODIS database[54]; the average Human Footprint Index, an integrated measure of human pressure in the environment[27]; and the average and maximum levels of artificial sky brightness[26]. Using a buffer distance of 1000 m allowed for obtaining data at the finest spatial resolution common to all mapped environmental indicators. Where investigated, the distances over which insects were attracted to artificial light sources did not exceed 1000 m (3–50 m in a review by ref. [55]; 10–519 m estimated by ref. [56]).

## Calculating and weighting effect sizes

We quantified the effect of diel period on the mean abundance or richness of insects observed using the natural log of the response ratio (lnRR), specifically the log proportional change in insect abundance or richness between night and day[57]. Even though data on SD were not reported in 55% ($n = 43$) of studies on the effect of diel period on insect abundance, we successfully calculated values of effect size ($y_i$) and sampling variance ($v_i$) based on the average between-study coefficient of variation. This method for estimating effect sizes, termed the 'all cases' method, has been shown to perform with minimal bias, regardless of the extent of missingness in the data, and has even shown to outperform the conventional approach of 'complete-case analysis' for estimating effect sizes and sampling variances from complete data (see ref. [58]). Nonetheless, we ran a sensitivity analysis for the meta-analysis on the effect of diel period on insect abundance by applying the 'complete-case analysis', and found that the results were qualitatively similar to the results of the meta-analysis using the 'all cases' method reported in the main text (see Supplementary Note 1). For the meta-analysis on the effect of diel period on insect richness, we used a complete-case analysis as SD were reported in all instances. In both meta-analyses, the effect sizes were weighted by the inverse of their sampling variance.

## Multi-level meta-analysis models

We ran separate meta-analyses to investigate the effects of diel period on insect abundance and richness. The 78 studies which met our inclusion criteria for the meta-analysis on insect abundance included a total of 331 effect sizes for day–night comparisons of insect abundance. The studies spanned all continents except Antarctica (Fig. 1), encompassed a wide range of habitats, including both terrestrial and aquatic insect communities, and sampled multiple major insect taxa (Supplementary Table 1). Several publications reported more than one effect size, with the different effect sizes corresponding to different sampling localities, sampling dates, or insect taxa sampled. These differences within- and between publications allowed us to investigate potential causes of variation (i.e. heterogeneity) in the abundance of active insects between day and night.

Including more than one effect size from the same publication, however, risks invalidating a meta-analytic model's assumptions of independence due to the correlation (clustering) of such effect sizes. We therefore used multi-level meta-analytic models to account for such dependency among effect sizes with random effects and sampling variance-covariance matrices[59]. These models follow the same principles as linear mixed effects models (LMMs; ref. [60]).

To estimate the overall effect of diel period (night vs. day) on the abundance of active insects, we built a 'null' multi-level meta-analytic model (RE.null) which included only random effects (i.e. analogous to an intercept-only LMM with no fixed effects). The random effects were a publication identifier (accounting for potential non-independence in clusters of effect sizes from the same study) and a unique effect size identifier (necessary to estimate residual heterogeneity). However, random effects alone do not control for the dependency arising from sampling errors that are shared among effect sizes (i.e. sampling error covariances)[59]. For example, if a publication reported multiple effect sizes because different insect taxa were sampled, the sampling errors of those effect sizes could co-vary if the different taxa were sampled collectively in the same event (i.e. they shared a common sampling locality and date). Therefore, besides modelling the dependency among effect sizes from clustering (via random effects), our meta-analytic model explicitly modelled sampling error co-variances using variance-covariance matrices[59] generated from locality-date clusters of the data. Here, we conservatively assumed a correlation (rho) value of 0.5 among sample variances of effect sizes obtained from the same sampling events. Nonetheless, we ran separate sensitivity analyses assuming either lower or higher correlation values of 0.1 and 0.9, respectively; the results were similar to those reported in the main text (Supplementary Note 1). We used $I^2$ (after ref. [61]) to estimate the total heterogeneity in the effect of diel period on insect abundance as well as the heterogeneity associated with different levels of clustering (random effects).

To investigate the effects of environmental factors on the heterogeneity in diel patterns in insect abundance, we added the variables of interest as moderators (fixed effects) in addition to the same random effects as in the RE.null model. These models were equivalent to LMMs with fixed and random effects. The moderators we tested and their hypothetical influence on diel activity patterns in insect communities are summarised in Supplementary Table 4. We first ran a separate model for each moderator, and established the importance of a moderator by comparing the AIC of its respective model to that of the null model (RE.null). We used marginal $R^2$ to measure the amount of heterogeneity explained by moderators[62]. All model comparisons were made using models fitted with maximum likelihood, while the results are reported from models fitted with restricted maximum likelihood. After identifying the individual moderators that outperformed the null model, we built a full model by conducting forward stepwise selection based on AIC. Specifically, we iteratively identified the best single-moderator model, followed by the best two-moderator model, then the best three-moderator model (and so on), until the inclusion of additional moderators did not result in an improved AIC score. Throughout this process, we excluded any models containing combinations of moderators that resulted in high collinearity (as indicated by a variance inflation factor exceeding 10).

We repeated the above steps of multi-level meta-analytic modelling to investigate the effect of diel period on insect richness with the comparatively limited data available on this variable (31 effect sizes from 13 studies). All data analysis was performed in R software version 4.3.0[63]. We used the metafor[64] package to build all meta-analytical models. We visualised the results from the models using scatterplots (for numeric moderators) and orchard plots (for categorical moderators) from the orchaRd[65] package, using 95% confidence intervals to indicate the most likely location of the cross-study average effect. We visualised the geographic distribution of studies using the rnaturalearth[66] package. We obtained silhouettes of insect orders to aid visualisation of the plots from PhyloPic[67].

## Reporting summary

Further information on research design is available in the Nature Portfolio Reporting Summary linked to this article.

## Data availability

All data generated in this study have been deposited at Figshare[68] with the identifier https://doi.org/10.6084/m9.figshare.24164652.

## Code availability

All code supporting the findings of this study have been deposited at Figshare[68] with the identifier https://doi.org/10.6084/m9.figshare.24164652.

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

## Acknowledgements

We wish to acknowledge the tireless contributions of the many entomologists working day and night to obtain crucial empirical data on insect communities. We are also grateful to Daniel Anderson for providing a template script for documenting the analysis. MKLW was supported by a Forrest Fellowship provided by the Forrest Research Foundation.

## Author contributions

M.K.L.W. conceived the idea and collected the data. M.K.L.W. conducted the analysis with advice from R.K.D. M.K.L.W. led the writing of the manuscript with substantial contributions from R.K.D.

## Competing interests

The authors declare no competing interests.
