## [Peer Review File · Nature Communications]

Global meta-analysis reveals overall higher nocturnal versus diurnal activity in insect communitiesReviewer #1 (Remarks to the Author):

This is an excellent study. It is asking a very simple question a child may ask, the answer to which is [i] important, [ii] unknown, and [iii] has been subject to various ecological hypotheses. The authors provide a convincing general answer that insects tend to be more active at night rather than during the day, but the value of the paper is especially in its detailed information on diurnal differences across insect taxa and ecological gradients, skillfully avoiding problems with inconsistencies between sampling methods, and allowing even some speculation on the effect of predation on insect abundance, not measured directly. I am not going to raise some minor point just for the sake of being able to suggest an improvement. Instead, this is one of the few reviews where I am happy with the first version of the paper.

Reviewer #2 (Remarks to the Author):

Review NCOMMS-23-51854-T
A third more insects out at night

The article "A third more insects out at night" by Wong and Didham is studying insect activity worldwide and disentangle the drivers of diel preferences found among a large number of taxa. This is a metadata paper gathering more than 300 quantitative comparisons across a range of contrasting habitat, taxonomic groups and various other environmental factors. If I have one main concern about the article is that the audience may not only be composed of entomologists so I recommend to the authors to add more details on the why we should study diel activity and preferences of major insect taxa. For instance, the paragraph starting line 37 may have more insights and informations, if possible, towards the reasons to study diel activity in insects.

Overall, I am amazed to only read such study in 2023 and I have to confess that I am surprising it did not exist in the literature. Hence, it is a novel, timely and very interesting study that I enjoyed reading. This is surely off great interest to the readers of Nature and I therefore recommend its publication with minor revision.

Some specific suggestions:

Line 29: I am too citing this Losez and Vaughan, 2006 for economic importance of insects, nothing more recent?

Line 37: What comes in mind at least for the species or taxa with nocturnal habits is the recent and timely threats of light pollution. I recommend the authors to (See <https://doi.org/10.1073/pnas.2023989118>). Some references can be found here: <https://resjournals.onlinelibrary.wiley.com/doi/10.1111/icad.12447>
<https://www.nature.com/articles/s41467-021-24394-0>
<https://resjournals.onlinelibrary.wiley.com/doi/10.1111/een.12174>

line 47 : "long timescale monitoring", see publications from our colleagues at <https://striresearch.si.edu/yves-basset-lab/>

line 57: I have to acknowledge that the study does include some tropical data which is great. However, since global arthropod diversity mostly occur in the tropics (see all work from Kitching, Stork, Basset, Novotny) I suggest the authors to mention at least that the pattern found may only reflect a small portion of the diversity. Don't you think?

Line 65-68: I am wondering if there is no reference at all on the how each methods (included in the

study) attract a different portion of nocturnal or diurnal taxa? For instance, a malaise or interaction traps would ultimately collect flying insects both at night and day.

Line 78: Do the authors think they should include the total number of insects in this paragraph with both the total abundance and species richness included in those 368 observations?

Figure 1: I am wondering if the authors have analysed the differences of diel pattern between temperate and tropical species? And if so, is it worth included this analysis in Figure 1c.

Line 133: puzzling that butterflies are higher at night? Can you please explain ?

Figure 2: Is the figure (and the mean lnRR) ranked in descending order? Such as Fig. 1B.

Line 160: I am personally puzzled by this result that the abundance of forest species was found lower during the night but even though if not significant, I recommend the author to dig on to this results and perhaps try to separate the "forest species" between temperate and tropical regions? It s only a suggestion but the diel pattern is surely interesting to check out across latitudes (Excel file Column E?)

Line 197-198: I suggest the authors starting here to state the mechanisms they will discuss to explain the pattern found and perhaps even add some sub headings into the next-to-come 5 paragraphs.

Line 197: "potentially" I will rephrase this part of the sentence since it's a bit vague.

Line 202: cf. Kearney et al 2009: nothing more recent than that? It is very interesting paragraph about thermal constraints. It will be excellent to go further in the discussion here. What about some works from <https://www.researchgate.net/profile/Raymond-Huey-2/research> or perhaps <https://jgking.web.unc.edu/publications/>

Line 237-245: converging comment than previously about the importance of tropical forest in overall diversity and what does this may imply to the pattern found by the authors? Also, may you name or enumerate the few mechanisms that may explain this great variation in diel activities? I am actually wandering now (coming after 1st full read) whether it will be interesting to mention if this difference may also be found between life stages (larvae vs. adult)?

Line 253: Ok I see in Table Supp that only 19 observations are from urban area. As I mentioned earlier for perspectives and future studies, it will be excellent to compare similar taxa between disturbed vs. pristine area.

Reviewer #1 (Remarks to the Author):

This is an excellent study. It is asking a very simple question a child may ask, the answer to which is [i] important, [ii] unknown, and [iii] has been subject to various ecological hypotheses. The authors provide a convincing general answer that insects tend to be more active at night rather than during the day, but the value of the paper is especially in its detailed information on diurnal differences across insect taxa and ecological gradients, skillfully avoiding problems with inconsistencies between sampling methods, and allowing even some speculation on the effect of predation on insect abundance, not measured directly. I am not going to raise some minor point just for the sake of being able to suggest an improvement. Instead, this is one of the few reviews where I am happy with the first version of the paper.

Response: We are delighted that the reviewer finds our study to be excellent and is happy with the first version of the manuscript. We are very grateful for the positive feedback on the content of the manuscript. We hope the reviewer will appreciate the efforts we have made to further improve the manuscript based on the comments of Reviewer 2 (below).

Reviewer #2 (Remarks to the Author):

Review NCOMMS-23-51854-T

A third more insects out at night

The article "A third more insects out at night" by Wong and Didham is studying insect activity worldwide and disentangle the drivers of diel preferences found among a large number of taxa. This is a metadata paper gathering more than 300 quantitative comparisons across a range of contrasting habitat, taxonomic groups and various other environmental factors.

Overall, I am amazed to only read such study in 2023 and I have to confess that I am surprising it did not exist in the literature. Hence, it is a novel, timely and very interesting study that I enjoyed reading. This is surely of great interest to the readers of Nature and I therefore recommend its publication with minor revision.

Response: We are delighted to hear that the reviewer agrees that insect diel activity patterns present a novel, timely and very interesting topic in ecological research. We are also very grateful for the suggested revisions, which have been very constructive and useful in improving the manuscript. We detail these changes in point-by-point responses to the individual comments below.

If I have one main concern about the article is that the audience may not only be composed of entomologists so I recommend to the authors to add more details on the why we should study diel activity and preferences of major insect taxa. For instance, the paragraph starting line 37 may have more insights and informations, if possible, towards the reasons to study diel activity in insects.

Line 37: What comes in mind at least for the species or taxa with nocturnal habits is the recent and timely threats of light pollution. I recommend the authors to

(See <https://doi.org/10.1073/pnas.2023989118>). Some references can be found here:
<https://resjournals.onlinelibrary.wiley.com/doi/10.1111/icad.12447>
<https://www.nature.com/articles/s41467-021-24394-0>
<https://resjournals.onlinelibrary.wiley.com/doi/10.1111/een.12174>

Response: We are grateful for this suggestion and the useful references provided. Accordingly, we have now expanded on our arguments for studying the diel activity patterns of insects, while adding new citations to the relevant references suggested by the reviewer.

L30: *“Insects are of tremendous ecological and economic importance because they sustain food webs, pollination networks and the functioning of ecosystems globally (Losey & Vaughan, 2006; Eggleton, 2020). However, many insect populations are currently threatened by human activities such as deforestation, agricultural intensification and urbanisation (Wagner et al., 2021).”*

L46: *“Investigating the diel activity patterns of insect communities is therefore not only vital for comprehending the temporal mechanisms determining community structure, but also the daily rhythms in key ecosystem functions. Such pursuits are timely considering mounting evidence of anthropogenic threats, such as light pollution, affecting specific communities and functions (Briolat et al., 2021; Dyer et al., 2023).”*

Some specific suggestions:

Line 29: I am too citing this Losey and Vaughan, 2006 for economic importance of insects, nothing more recent?

Response: As far as we know, the paper by Losey and Vaughan (2006) remains the most authoritative study that reports a monetary estimate – albeit only for the United States – of the economic value of ecosystem services provided by insects. It appears that a future update on this study and expansion to the global scale would be an important and timely pursuit!

line 47 : “long timescale monitoring”, see publications from our colleagues at <https://stiresearch.si.edu/yves-basset-lab/>

Response: We are grateful for the link to find additional studies that are relevant here. In the revised manuscript, we have cited the useful review on the topic of long-term monitoring of tropical insects by Lamarre and colleagues (2020), as well as the recent excellent study on long-term monitoring of tropical social insect assemblages by Basset and colleagues (2023).

Basset, Y., Butterill, P. T., Donoso, D. A., Lamarre, G. P., Souto-Vilarós, D., Perez, F., ... & Barrios, H. (2023). Abundance, occurrence and time series: long-term monitoring of social insects in a tropical rainforest. *Ecological Indicators*, 150, 110243.

Lamarre, G.P.A., Fayle T.M., Segar, S.T., Laird-Hopkins, B., Nakamura, A., Souto-Vilarós, D., Watanabe, S. & Basset, Y. 2020. Monitoring tropical insects in the 21st century. *Advances in Ecological Research*, 62, 295-330.

line 57: I have to acknowledge that the study does include some tropical data which is great. However, since global arthropod diversity mostly occur in the tropics (see all work from Kitching, Stork, Basset, Novotny) I suggest the authors to mention at least that the pattern found may only reflect a small portion of the diversity. Don't you think?

Response: We agree with the reviewer that this is an important point to highlight, and we have now added a sentence in the Introduction so that readers can better interpret our results within this context.

L84: *"Nevertheless, as with most global biodiversity assessments (Hughes et al., 2021), tropical regions remained underrepresented relative to the high concentrations of biodiversity in these areas (Novotny et al., 2006; Basset et al., 2012)."*

Line 65-68: I am wondering if there is no reference at all on the how each methods (included in the study) attract a different portion of nocturnal or diurnal taxa? For instance, a malaise or interaction traps would ultimately collect flying insects both at night and day.

Response: Kindly allow us to clarify this issue. Indeed, in most studies that do not specifically investigate diel activity patterns, the collection jars of malaise traps, flight interception traps and pitfall traps are typically left exposed for periods exceeding 12 hours, resulting in the collection of both day- and night-active insects in the same sample.

However, our metaanalysis focused on studies that specifically investigated the diel activity patterns of insect communities. As such, we only included a study in our metaanalysis if the investigators used the same sampling method during the day and the night, and separated day collections from night collections.

That is, the investigators would set up their malaise-, flight interception-, or pitfall traps after sunrise, then return before sunset to retrieve the jars containing the insects collected during the day. They would then place new (i.e., empty) jars at the base of the traps to contain insects collected during the night. The following day, they would return before sunrise to retrieve the jars of insects collected during the night. Depending on the study, this sampling protocol may have been repeated over several days and seasons, but day and night samples were always separated. A few examples of such studies included in our metaanalysis are listed further below.

We have revised a sentence in our Methods to more clearly reflect these inclusion criteria in our metaanalysis.

L355: *"We only included a study in our metaanalysis if it systematically sampled an insect community using the same collection method during the day and the night, and separated the samples collected from each diel period."*

Such approaches to sampling insect communities are very labour intensive, but necessary to elucidate diel activity patterns. Hence, in our Acknowledgements, we express our gratitude to entomologists that have undertaken such important and challenging empirical work.

L477: *“The authors wish to acknowledge the tireless contributions of the many entomologists working day and night to obtain crucial empirical data on insect communities.”*

Example studies:

Dudgeon, D. (2006). The impacts of human disturbance on stream benthic invertebrates and their drift in North Sulawesi, Indonesia. *Freshwater Biology*, 51(9), 1710-1729.

Lieberman, S., & Dock, C. F. (1982). Analysis of the leaf litter arthropod fauna of a lowland tropical evergreen forest site (La Selva, Costa Rica). *Revista de Biología Tropical*, 30(1), 27-34.

Springate, N. D., & Basset, Y. (1996). Diel activity of arboreal arthropods associated with Papua New Guinean trees. *Journal of Natural History*, 30(1), 101-112.

Line 78: Do the authors think they should include the total number of insects in this paragraph with both the total abundance and species richness included in those 386 observations?

Response: We have estimated the number of individuals sampled across the 331 observations in our metaanalysis for the diel effects on abundances, and reported this in the manuscript as recommended.

L86: *“The insects sampled across all studies exceeded 3 million individuals from 16 taxonomic orders...”*

It is important to note that this is a minimum estimate, because it excludes 22 studies that measured abundance in non-compatible formats, e.g., in terms of biomass (in grams) or in terms of frequencies of occurrence (e.g., % of total individuals / baits / biomass).

We agree that it would be ideal to report the total species richness captured across all observations compiled from the literature. Unfortunately, the incomplete nature of the data precluded us from determining the total species richness captured across all observations. To do so, we would require the full list of species recorded in each observation. This would allow us to check for duplicated species and taxonomic consistencies across observations, and thereby determine the total number of species observed. However, many studies, especially those sampling multiple major taxonomic groups, did not identify the insects to the species level, or state the total species richness, let alone provide a full list of the species observed. We discuss this limitation in the taxonomic resolution of the literature in our manuscript:

L88: “However, likely owing to taxonomic impediments, very few studies reported the mean richness of insect taxa observed during a specific diel period...”

L298: “In particular, the diel partitioning of species richness in insect communities is largely unexplored, and it remains to be seen whether diel patterns in species richness mirror those of abundances across insect communities globally (or if these indeed vary along other environmental gradients such as latitude; see Supplementary Fig. 1). To this end, increasingly accessible molecular methods for sorting specimens such as DNA barcoding (Chua et al., 2023) offer promising avenues for overcoming the taxonomic impediments that have hindered studies on the diel dynamics of insect communities in diverse regions, particularly tropical ecosystems.”

Line 133: puzzling that butterflies are higher at night? Can you please explain ?

Response: To clarify, this result reports the diel abundance pattern for the order Lepidoptera as a whole, including both moths and butterflies. We were unable to separate moths and butterflies in our meta-analysis because the original studies in the literature did not do so. Given that moths have higher species richness and abundances than butterflies in most locations (e.g., Ricketts et al., 2002), and considering that many moth species are nocturnal, it is perhaps unsurprising that the overall abundances of Lepidoptera were higher at night on average.

Ricketts, T. H., Daily, G. C., & Ehrlich, P. R. (2002). Does butterfly diversity predict moth diversity? Testing a popular indicator taxon at local scales. *Biological conservation*, 103(3), 361-370.

Figure 2: Is the figure (and the mean lnRR) ranked in descending order? Such as Fig. 1B.

Response: Thank you for pointing this out. For clarity and consistency, we have since remade all the caterpillar plots and orchard plots (Fig. 1b, Fig. 2 and Fig. 3 in the revised manuscript) such that the individual studies (Fig. 1b) or factor levels (Fig. 2 and Fig. 3) are consistently presented in the order of ascending mean effect sizes (i.e., decreasing diurnality and increasing nocturnality) from top to bottom. We have also added a line to each respective figure legend to clarify this order of presentation, e.g., “Taxa are presented in the order of ascending mean effect sizes from top to bottom.”

Figure 1: I am wondering if the authors have analysed the differences of diel pattern between temperate and tropical species? And if so, is it worth included this analysis in Figure 1c.

Response: We appreciate this useful suggestion. We have since run a single-moderator model (‘RE.troptemp’) to investigate the potential moderating effect of geographic region (2 levels: “tropical” and “temperate”) on diel abundance patterns. We found that this model did not outperform the null (intercept-only) model, (RE.troptemp AIC=1090.00 vs. RE.null AIC=1088.20), suggesting that in our data, geographic region did not have a strong moderating effect on diel abundance patterns. Nevertheless, we have included the results of this single-moderator model (RE.troptemp) in Supplementary Table 1.

In our original analysis, we also investigated the effect of latitude (measured as the absolute latitude from the equator) in moderating diel abundance patterns. The single-moderator model for this factor did not outperform the null model (Table S1: RE.latitude AIC=1089.09, vs. RE.null AIC=1088.2).

We can empathise with the referee's sentiments, because as entomologists we too would intuitively expect tropical regions to have high nocturnal insect activity (based on our field experiences). However, this was apparently not supported by the current data, where latitude and geographic region (tropical vs. temperate) did not have strong moderating effects on diel abundance patterns in our dataset. Apparently, this is also not supported by some of the primary literature. For instance, in a well documented and rigorous sampling of the diel activity patterns of a tropical insect community in Papua New Guinea by Springate and Basset (1994), the authors found that *“Arthropod activity (measured by the number of individuals collected) was correlated positively with minimum air temperature. As a probable consequence, overall arthropod activity was significantly higher during day-time than night-time and night catches represented 72% of day catches.”*

That being said, it is worth noting that in our limited analysis for diel effects on species richness (with 31 effect sizes), the effect of latitude showed a strong moderating effect (see Supplementary Note 1), with tropical regions being associated with higher nocturnal species richness overall (Supplementary Fig. 1). Nonetheless, as this analysis was performed with very limited data (compared to our analysis on diel abundance patterns), we intentionally refrained from speculating on the ecological interpretation of these results. We believe that more studies on the diel partitioning of species richness are needed to clarify the existence of a latitudinal gradient in diel patterns of species richness in insect communities. We describe this in our discussion:

L297: *“In showing that insect abundances vary across the diel cycle, our findings shed light on a general and but still poorly understood ecological phenomenon. In particular, the diel partitioning of species richness in insect communities is largely unexplored, and it remains to be seen whether diel patterns in species richness mirror those of abundances across insect communities globally (or if these indeed vary along other environmental gradients such as latitude; see Supplementary Fig. 1). To this end, increasingly accessible molecular methods for sorting specimens such as DNA barcoding (Chua et al., 2023) offer promising avenues for overcoming the taxonomic impediments that have hindered studies on the diel dynamics of insect communities in diverse regions, particularly tropical ecosystems.”*

Line 160: I am personally puzzled by this result that the abundance of forest species was found lower during the night but even though if not significant, I recommend the author to dig on to this results and perhaps try to separate the “forest species” between temperate and tropical regions? It s only a suggestion but the diel pattern is surely interesting to check out across latitudes (Excel file Column E?)

Response: As detailed in our response to the above comment, we did not detect strong moderating effects of geographic region (tropical vs. temperate) or latitude on diel abundance patterns. As such, we had no justification to include these moderators in the

multi-moderator model. Nonetheless, as suggested by the referee, we have plotted the data to explore if there might be different effects of diel period on insect abundance for forests in tropical versus temperate regions. As shown in the exploratory plot below, the effect sizes for forest insect communities in both tropical and temperate regions show relatively similar distributions. This implies that the effect of diel period (night vs. day) on insect abundance in forests was not strongly influenced by geography (i.e., whether a forest was in a tropical or temperate region), at least based on our current data. That being said, it is worth noting that in our limited analysis for diel effects on species richness, the effect of latitude showed a strong moderating effect, and we report this in the Supplementary Information of our manuscript (kindly refer to our response to comment above).

Line 237-245: converging comment than previously about the importance of tropical forest in overall diversity and what does this may imply to the pattern found by the authors? Also, may you name or enumerate the few mechanisms that may explain this great variation in diel activities?

Response: As outlined in our response to an earlier comment, we agree with the reviewer that the underrepresentation of tropical areas is an important point to highlight, and have added a sentence in the Introduction so that readers can better interpret our results within this context.

L84: “Nevertheless, as with most global biodiversity assessments (Hughes et al., 2021), tropical regions remained underrepresented relative to the high concentrations of biodiversity in these areas (Novotny et al., 2006; Basset et al., 2012).”

We initially opted to avoid conclusively stating the individual ecological mechanisms in the first paragraph of the discussion because they remain somewhat speculative. We have since added a sentence to list a few candidate mechanisms which we discuss, while acknowledging that they are not exhaustive.

L213: *“Below, we discuss a few of these potential mechanisms (e.g., abiotic regulation, predator avoidance, resource tracking, intraspecific variation), noting however that others are likely also at play, and their specific roles in structuring the diel dynamics of insect communities will best be illuminated through additional field and experimental studies across the 24-hour diel cycle.”*

I am actually wandering now (coming after 1st full read) whether it will be interesting to mention if this difference may also be found between life stages (larvae vs. adult)?

Response: This is a great point and something that we are curious about as well. We have added a paragraph in our revised Discussion addressing this issue and proposing a hypothesis for future work to address.

L268: *“It is also possible that the variation in diel activity patterns observed across insect communities is shaped to some degree by intraspecific variation in the activity patterns of individuals from different stages across the life cycle. Although some studies in our analysis only sampled adults (e.g., studies on ants, wasps and ground beetles), others targeted juveniles (e.g., studies on the drift of chironomid larvae in streams), and many used methods that may have collected individuals from different life stages without distinguishing these in samples (e.g., pitfall traps and drift nets may collect both juvenile and adult insects in terrestrial and aquatic ecosystems, respectively). Yet in many insects, juveniles differ markedly from adults in ways that should influence their periods of activity across the diel cycle, such as in their specific diets and habitats, mobility and foraging patterns, and susceptibility to predators (Nakazawa, 2015; Sporer et al., 2020). Moreover, juvenile and adult insects can possess different thermal sensitivities, with the thermal tolerances of some species either increasing or decreasing following developmental change (Kingsolver & Buckley, 2020). While we suspect that nocturnal insect communities in aquatic and terrestrial systems may be disproportionately comprised of juvenile individuals avoiding predators and higher temperatures, this remains to be tested empirically.”*

Kingsolver, J. G., & Buckley, L. B. (2020). Ontogenetic variation in thermal sensitivity shapes insect ecological responses to climate change. *Current Opinion in Insect Science*, 41, 17-24.

Nakazawa, T. (2015). Ontogenetic niche shifts matter in community ecology: a review and future perspectives. *Population Ecology*, 57(2), 347-354.

Sporer, T., Körnig, J., & Beran, F. (2020). Ontogenetic differences in the chemical defence of flea beetles influence their predation risk. *Functional Ecology*, 34(7), 1370-1379.

Line 197-198: I suggest the authors starting here to state the mechanisms they will discuss to explain the pattern found and perhaps even add some sub headings into the next-to-come 5 paragraphs.

Response: We initially opted to avoid conclusively stating the individual ecological mechanisms in the first paragraph of the discussion because they remain somewhat speculative. We have since added a sentence to list a few candidate mechanisms which we discuss, while acknowledging that they are not exhaustive.

L213: *“Below, we discuss a few of these potential mechanisms (e.g., abiotic regulation, predator avoidance, resource tracking, intraspecific variation), noting however that others are likely also at play, and their specific roles in structuring the diel dynamics of insect communities will best be illuminated through additional field and experimental studies across the 24-hour diel cycle.”*

We have not been able to add subheadings as *Nature Communications* does not permit the use of subheadings in the Discussion section of research articles.

Line 197: “potentially” I will rephrase this part of the sentence since it’s a bit vague.

Response: Thank you for this suggestion. We revised the sentence to improve its clarity. In particular, we have replaced the phrase “potentially” with “likely” to make a clearer and stronger statement.

L209: *“Moreover, in uncovering multiple key moderators of this variation—which include a range of abiotic factors (climate, elevation) and biotic factors (e.g., habitat type, taxonomic constitution, productivity gradients)—our results underscore that the mechanisms governing the diel dynamics of insect community structure are likely rich and varied.”*

Line 202: cf. Kearney et al 2009: nothing more recent than that? It is very interesting paragraph about thermal constraints. It will be excellent to go further in the discussion here. What about some works from <https://www.researchgate.net/profile/Raymond-Huey-2/research> or perhaps <https://jgking.web.unc.edu/publications/>

Response: We have replaced Kearney et al. (2009) with a more recent and relevant reference, González-Tokman et al. (2020), which reviews the topic of insect responses to heat. We are grateful for the recommendation to the very interesting work of Huey and Kingsolver. With citations to a few of their relevant papers, we have expanded our explanation on the role of thermal constraints in shaping diel patterns, the different thermal sensitivities of adult and juvenile insects, and the threat that climate warming poses to nocturnal communities in particular.

L221: *“The effects of temperature and elevation in shaping the distribution of insect abundance across day and night periods in communities worldwide are consistent with the notion that the activities of these ectothermic organisms are broadly thermally constrained (González-Tokman et al., 2020). Thermal Performance Theory (Kingsolver & Huey 2008) posits that the physiological performance of ectotherms increases with temperature until reaching a peak, beyond which there is a rapid decline in physiological performance, ultimately leading to the ectotherm’s death. Given that environmental temperatures peak during the day, higher maximum environmental temperatures may select for increased*

nocturnality in insect communities (Fig. 4a) as more individuals avoid heat stress from daytime temperatures that approximate their upper thermal limits (Hoffmann et al., 2013)."

L278: *"Moreover, juvenile and adult insects can possess different thermal sensitivities, with the thermal tolerances of some species either increasing or decreasing following developmental change (Kingsolver & Buckley, 2020)."*

L318: *"...climate warming may further shorten periods of biophysically feasible activity for nocturnal insects in warm environments such as tropical forests, with complex consequences for fitness such as accelerated metabolic costs combined with reduced food intake (Speights et al., 2017; Huey & Kingsolver, 2019)."*

González-Tokman, D., Córdoba-Aguilar, A., Dáttilo, W., Lira-Noriega, A., Sánchez-Guillén, R. A., & Villalobos, F. (2020). Insect responses to heat: physiological mechanisms, evolution and ecological implications in a warming world. *Biological Reviews*, 95(3), 802-821.

Kingsolver, J., & Huey, R. (2008). Size, temperature, and fitness: three rules. *Evolutionary Ecology Research*, 10(2), 251-268.

Kingsolver, J. G., & Buckley, L. B. (2020). Ontogenetic variation in thermal sensitivity shapes insect ecological responses to climate change. *Current Opinion in Insect Science*, 41, 17-24.

Line 253: Ok I see in Table Supp that only 19 observations are from urban area. As I mentioned earlier for perspectives and future studies, it will be excellent to compare similar taxa between disturbed vs. pristine area.

Response: We agree that investigating the effects of anthropogenic disturbances on the diel patterns of insect communities is an important area for future research, and emphasise this in new sentences added to the revised manuscript:

L293: *"To this end, additional studies on the diel activity patterns of insect populations and communities along disturbance gradients may clarify the mechanisms driving human impacts on insect biodiversity."*

[Editor's note: Reviewer #1 was not contacted; Reviewer #2 had no further comments]

Reviewer #3 (Remarks to the Author):

I am grateful for the opportunity to comment on the methodology of this study, which addresses a very interesting and important question, integrating an impressive array of studies from around the world to document diel activity patterns of insect communities.

My primary critique is that the title of the paper, and main takeaway ("A third more insects out at night"), are a bit misleading. At a 95% confidence level, the results suggest that the true mean difference in insect abundance from the day to the night ranges somewhere from -6% to +84%. While the center of this range is 31.4%, this is a large confidence interval that includes 0, and so I'm not sure that we can conclude all that much about whether insects are more active during the night relative to the day as a general rule. It strikes me that the main take-away is that there is, in fact, widespread variation in whether insects are more active in the day vs. night, across systems, rather than there being a consistent global trend. This may require some reframing.

Otherwise, the meta-analytical methods themselves appear sound. The authors use the best practices and scripts presented by the recent manuscript by Nakagawa et al. (2023). The use of random effects and modeling co-variance in sampling error to account for multiple effect sizes for many studies was appropriate, and the associated sensitivity analysis satisfactory. There were no data on sampling variance for 55% of the studies, which is quite high, but the sensitivity analysis is satisfactory, and the estimates of variance for those studies missing information is conservative.

Some questions/comments:

There is a lot of great commentary, explanation of methods, and interpretation of results in the R script, some of which is missing from the manuscript/supplement. I would ensure that everything in the script is also somewhere in the manuscript or supplement, as most readers will never open the script.

For example, the model selection process was not clearly explained (outside of the R script). What combinations of covariates were tested? Was it a stepwise model building process? Also, nowhere are the tested covariates justified—what were the underlying hypotheses about how each of them might affect diel activity? This information could perhaps be included in supplementary material, if there is inadequate space in the main text.

A minor thing about Figures 2 and 3: the size of the circles is meant to scale with precision ($1/SE$), but there is very little discrepancy in the size (I cannot distinguish any differences with the naked eye). I recommend either increasing the variation in sizes, or just making them all the same. Given that this metric was estimated for over half of the studies, perhaps it doesn't make sense to highlight it in the figure.

Line 349: How many references did the search return? The authors mention screening the first 2,000, but it is unclear of the total amount.

[Editor's note: Reviewer #1 was not contacted; Reviewer #2 had no further comments]

Reviewer #3 (Remarks to the Author):

I am grateful for the opportunity to comment on the methodology of this study, which addresses a very interesting and important question, integrating an impressive array of studies from around the world to document diel activity patterns of insect communities.

My primary critique is that the title of the paper, and main takeaway (“A third more insects out at night”), are a bit misleading. At a 95% confidence level, the results suggest that the true mean difference in insect abundance from the day to the night ranges somewhere from -6% to +84%. While the center of this range is 31.4%, this is a large confidence interval that includes 0, and so I’m not sure that we can conclude all that much about whether insects are more active during the night relative to the day as a general rule. It strikes me that the main take-away is that there is, in fact, widespread variation in whether insects are more active in the day vs. night, across systems, rather than there being a consistent global trend. This may require some reframing.

Response: We agree with the reviewer that the original title and several parts of the manuscript failed to appropriately account for the important and extensive variation in diel activity patterns that we found among insect communities globally. Accordingly, we have replaced the title, and revised sections of the Abstract and Results to emphasise that the overall effect is subject to extensive variation.

New title to explicitly state that the general finding is an ‘overall’ effect size:

“Global meta-analysis reveals overall higher nocturnal versus diurnal activity in insect communities”

New text added to Abstract at the recommendation of the Editor to emphasise the variation (in bold):

*“Here, we assemble 331 quantitative comparisons of the abundances of insects between day and night periods from 78 studies worldwide and use multi-level meta-analytical models to show that insect activity is on average 31.4% (CI: -6.3%–84.3%) higher at night than in the day... **The highly variable but overall high** nocturnal activity in insect communities underscores...”*

New text added to Results to emphasise the variation (in bold):

*“Using multi-level meta-analysis modelling, the abundance of insects was found to be higher in the night than in the day by an average of 31.4%, **but this relationship varied considerably** (CI: -6.3%–84.3%) across the 331 observations from 78 publications...”*

Otherwise, the meta-analytical methods themselves appear sound. The authors use the best practices and scripts presented by the recent manuscript by Nakagawa et al. (2023). The use of random effects and modeling co-variance in sampling error to account for multiple effect sizes for many studies was appropriate, and the associated sensitivity analysis satisfactory. There were no data on sampling variance for 55% of the studies, which is quite high, but the sensitivity analysis is satisfactory, and the estimates of variance for those studies missing information is conservative.

Response: We appreciate the reviewer's assessment of the meta-analytical approach and associated sensitivity analyses.

Some questions/comments:

There is a lot of great commentary, explanation of methods, and interpretation of results in the R script, some of which is missing from the manuscript/supplement. I would ensure that everything in the script is also somewhere in the manuscript or supplement, as most readers will never open the script.

For example, the model selection process was not clearly explained (outside of the R script). What combinations of covariates were tested? Was it a stepwise model building process? Also, nowhere are the tested covariates justified—what were the underlying hypotheses about how each of them might affect diel activity? This information could perhaps be included in supplementary material, if there is inadequate space in the main text.

Response: We thank the reviewer for their compliments on the contents of our R script, which we prepared and annotated in detail to facilitate the reproducibility of the research. We agree that since most readers would probably not explore the R script, it would be best to reflect its contents in the main article or supplementary information file. Upon comparing the contents of our R Script and the article files, we determined that the content that was missing from the article was a detailed description of the model selection and building process, as well as the justification of the choice of environmental variables that were tested as moderators in the meta-analysis. Both issues were identified by the reviewer, and we address them below.

We have expanded on the text in our Methods to describe our model selection process in greater detail, including the process of building a multi-moderator model.

L479: *"To investigate the effects of environmental factors on the heterogeneity in diel patterns in insect abundance, we added the variables of interest as moderators (fixed effects) in addition to the same random effects as in the RE.null model. These models were equivalent to LMMs with fixed and random effects. The moderators we tested and their hypothetical influence on diel activity patterns in insect communities are summarised in Supplementary Table 4. We first ran a separate model for each moderator, and established the importance of a moderator by comparing the AIC of its respective model to that of the null model (RE.null). We used marginal R^2 to measure the amount of heterogeneity explained by moderators⁶³. All model comparisons were made using models fitted with maximum likelihood, while the results are reported from models fitted with restricted maximum*

likelihood. After identifying the individual moderators that outperformed the null model, we built a full model by conducting forward stepwise selection based on AIC. Specifically, we iteratively identified the best single-moderator model, followed by the best two-moderator model, then the best three-moderator model (and so on), until the inclusion of additional moderators did not result in an improved AIC score. Throughout this process, we excluded any models containing combinations of moderators that resulted in high collinearity (as indicated by a variance inflation factor exceeding 10)."

We have also added a new table, Supplementary Table 4, which presents the environmental variables included in the meta-analysis, describes how data for each was collected and classified, and presents the justification for including them in the meta-analysis on diel activity patterns. We also refer readers to this table in the main text.

A minor thing about Figures 2 and 3: the size of the circles is meant to scale with precision (1/SE), but there is very little discrepancy in the size (I cannot distinguish any differences with the naked eye). I recommend either increasing the variation in sizes, or just making them all the same. Given that this metric was estimated for over half of the studies, perhaps it doesn't make sense to highlight it in the figure.

Response: We thank the reviewer for this suggestion. We have followed the best practices (Nakagawa et al. 2021; 2023) in creating the orchard plots in Figures 2 and 3. A key feature of the orchard plot which is the display of any variation in the precision of effect size values (Nakagawa et al. 2021; 2023). We created the plots using the function "orchard_plot" of the *orchaRd* R package (Nakagawa et al. 2023). The function has a built-in argument to automatically scale the sizes of circles indicating effect sizes according to their respective precision. While we agree that it takes some effort to see this variation in precision in the figures, we would prefer to retain it (rather than exclude it) for data transparency, and also because the illustration of any variation in precision is an integral feature of orchard plots (Nakagawa et al. 2021; 2023). Furthermore, we note that in Nakagawa et al. (2023), multiple examples of orchard plots are presented (Figures 1–4 in the article), and in many of these the variation in the precision values is not always obvious from the differences in circle sizes; however this variation is always kept in the plots (i.e. the sizes of the circles they present are never made to be exactly the same).

Nakagawa, S., Lagisz, M., O'Dea, R. E., Rutkowska, J., Yang, Y. F., Noble, D. W. A., & Senior, A. M. (2021). The orchard plot: Cultivating a forest plot for use in ecology, evolution, and beyond. *Research Synthesis Methods*, 12, 4–12.

Nakagawa, S., Lagisz, M., O'Dea, R. E., Pottier, P., Rutkowska, J., Senior, A. M., ... & Noble, D. W. (2023). *orchaRd* 2.0: an R package for visualizing meta-analyses with orchard plots. <https://besjournals.onlinelibrary.wiley.com/doi/10.1111/2041-210X.14152>

Line 349: How many references did the search return? The authors mention screening the first 2,000, but it is unclear of the total amount.

Response: We have now provided the details of the number of results returned from the search. These details are also shown in the PRISMA diagram, which we now include as Supplementary Fig. 3 in the Supplementary Information file.

L378: *“We sorted the 19,983 results by relevance and manually screened the abstracts of the first 2000 results to identify relevant publications (beyond the 1350th result, zero to two relevant publications were identified out of every 50 results, while no relevant publications were identified beyond the 1850th result).”*